# Study on P-AlGaAs/Al/Au Ohmic Contact Characteristics for Improving Optoelectronic Response of Infrared Light-Emitting Device

**DOI:** 10.3390/mi14051053

**Published:** 2023-05-16

**Authors:** Hyung-Joo Lee, Jae-Sam Shim, Jin-Young Park, Lee-Ku Kwac, Chang-Ho Seo

**Affiliations:** 1CF Technical Division, AUK Corporation, Iksan 54630, Republic of Korea; dieblood77@nate.com; 2Korea Photonic Technology Institute (KOPTI), Gwangju 61007, Republic of Korea; jss9696@kopti.re.kr (J.-S.S.); police7630@kopti.re.kr (J.-Y.P.); 3Graduate School of Carbon Convergence Engineering, Jeonju University, Jeonju 55069, Republic of Korea; 4Department of Convergence Science, Kongju National University, Chungnam 32588, Republic of Korea

**Keywords:** Al/Au alloy, p-AlGaAs, ohmic contact, reflective structure, light-emitting diode

## Abstract

The Al/Au alloy was investigated to improve the ohmic characteristic and light efficiency of reflective infrared light-emitting diodes (IR-LEDs). The Al/Au alloy, which was fabricated by combining 10% aluminum and 90% gold, led to considerably improved conductivity on the top layer of p-AlGaAs of the reflective IR-LEDs. In the wafer bond process required for fabricating the reflective IR-LED, the Al/Au alloy, which has filled the hole patterns in Si_3_N_4_ film, was used for improving the reflectivity of the Ag reflector and was bonded directly to the top layer of p-AlGaAs on the epitaxial wafer. Based on current-voltage measurements, it was found that the Al/Au alloyed material has a distinct ohmic characteristic pertaining to the p-AlGaAs layer compared with those of the Au/Be alloy material. Therefore, the Al/Au alloy may constitute one of the favored approaches for overcoming the insulative reflective structures of reflective IR-LEDs. For a current density of 200 mA, a lower forward voltage (1.56 V) was observed from the wafer bond IR-LED chip made with the Al/Au alloy; this voltage was remarkably lower in value than that of the conventional chip made with the Au/Be metal (2.29 V). A higher output power (182 mW) was observed from the reflective IR-LEDs made with the Al/Au alloy, thus displaying an increase of 64% compared with those made with the Au/Be alloy (111 mW).

## 1. Introduction

Near-infrared light-emitting diodes (NIR-LEDs) were recently used as emitters for optical sensors in wearable devices, small vehicles, time-of-flight (T.O.F) sensors, and flying drones [1,2]. To meet the demands of these applications, small NIR-LEDs that can deliver higher power outputs at large injection currents are required. For several years, multiple quantum wells (MQWs), distributed Bragg reflector (DBR), omni-directional reflector (ODR), and current spreading layers have been used to improve the output powers in NIR-LEDs [3,4,5,6]. However, the abrupt decrease in the surface-emitting area of these devices, due to a reduction in the chip size, was not investigated in prior research efforts [7,8,9,10].

Based on previous studies, it could be assumed that the use of ODR by the wafer bonding process is an effective approach for the remarkable improvement of the light efficiency of LEDs [11,12,13,14]. Most light photons, which are emitted from the active region to the absorbing substrate, are effectively reflected in either the upper surface or sidewalls by the reflective function of ODR. It is known that the reflectivity of the reflector is essentially influenced by the metal and structural frame of the reflective structure used as ODR. Therefore, reflective structures used for the ODR are very important for effective improvements of the output power of NIR-LEDs. Reflective structures typically consist of both reflectors and insulators. Herein, an insulator has an important role in sustaining the reflectivity of the reflector. This insulator, which is inserted between the Si substrate and the LED, clearly increases the series resistance in the reflective LED. Therefore, the insulator has been universally patterned for use in reflective LEDs. The conductive characteristic of ohmic materials on patterned insulators should be studied to reduce the series resistance of reflective LEDs.

In addition, the top layer of the IR-LED (in contact with the ohmic material) should also be considered to improve ohmic contact and reflectivity. Previously, the GaP top layer was mainly used because of the existent Au/Be ohmic alloy material and non-absorption surface. However, the GaP, which causes a lattice mismatch (3.57 %) to (Al)GaAs, is no more useful for reflective (Al)GaAs based IR-LEDs owing to the use of the insulators for reflective structures [15]. A lattice mismatch between Al_x_Ga_1−x_As (x = 0) and Al_x_Ga_1−x_As(x = 1) is known as approximately under 0.5 % [16]. 

In this manuscript, we would like to use the p-AlGaAs contact layer to obtain 850 nm IR-LED with higher power due to the p-GaP contact layer, which is one of the absorbing factors on lights emitted from the active region. The GaP (5.45 Å) is lattice-mismatched with Al_x_Ga_1−x_As(~5.62 Å) mainly used for 850 nm IR-LEDs [17]. Therefore, there are a lot of lattice defects between GaP and AlGaAs materials. Light in the LED structure was absorbed and reduced by lattice defects.

In this study, we investigated the Al/Au alloyed material in an effort to effectively improve the characteristics of ohmic contacts in reflective IR-LEDs made with a p-AlGaAs top layer. The Al/Au alloyed material with a 10% Al composition was the favorite for improving the light efficiency and series resistance of reflective NIR-LEDs.

## 2. Materials and Methods

The Al_x_Ga_1−x_As layers used were grown on an *n*-type GaAs (111) substrate at a 10 tilt toward [0−1-1], using a metal organic chemical vapor deposition (MOCVD) system. The GaAs substrate used was lattice-matched with Al_x_Ga_1−x_As. Trimethylgallium (TMGa), trimethyl-aluminum (TMAl), and trimethyl-indium (TMIn) were used as group III sources, and arsine (AsH_3_), and phosphine (PH_3_) were used as group V sources. Disilane (Si_2_H_6_) gas and cyclopentadienyl magnesium (Cp2Mg) were used as the *n* and p-dopant sources, respectively. Hydrogen (H_2_) gas was used as the source carrier.

Generally, various multiple quantum well (MQW) (active region) structures of different wavelengths are located between the *n* and *p* confinement layers. Four pairs of MQWs, each with 5 nm-thick GaAs wells and 12 nm-thick Al_0.05_Ga_0.95_As barriers, were used as the active region. The active region was located between the *n*- and *p*-type confinement layers, which are *n*- and *p-doped* Al_0.3_Ga_0.7_As materials, respectively. The photonic excitations of the MQWs were measured using a photoluminescence (P.L.) system with laser sources of 660 nm. The MQW structure of 850 nm was measured with a laser source of 660 nm. Normally, the epitaxial layer (active region) was analyzed through using the TEM system. Figure 1 shows the TEM image and PL data for the developed 850 nm MQW (active region).

For the wafer bond process, n-GaInP and 5 µm-thick n-Al_0.2_Ga_0.8_As layers, grown sequentially on the GaAs substrate, were employed as the current-spreading effect and etching-stop layer (ESL), respectively. Developed reflective structures with different optical thicknesses and repeated 10 µm-hole patterns were fabricated on the n-Al_0.3_Ga_0.7_As layer.

A 110 nm-thick Si_3_N_4_ (*n* = 2.0 @ 850 nm) film was deposited on the *p-doped* Al_0.3_Ga_0.7_As layer to maintain reflectivity of the Ag reflector. The Si_3_N_4_ film on epitaxial wafer was coated and patterned by a photo-lithography process. The Si_3_N_4_ with photo-resistance hole patterns was selectively etched by plasma etching through using an RIE etcher with CF4 gas. The epitaxial wafer with patterned PR/Si_3_N_4_ film was dipped in BOE solution for 5 s. It was loaded on evaporation to deposit the Al_0.1_Au_0.9_ alloy. After the lift-off process, the Al_0.1_Au_0.9_ alloy was filled selectively into the vacant area (*p-doped* Al_0.3_Ga_0.7_As layer) of Si_3_N_4_ film.

Developed reflective structures, which have different optical thicknesses and repeated 10 μm hole patterns, were fabricated on the Si_3_N_4_ (*n* = 2.0 @ 850 nm). Approximately 80 patterns exist with a diameter of 10 μm in one fabricated 14 mil NIR-LED chip. For bonding wafer to wafer, a 400 nm-thick Ag material and 5000 nm-thick Ti/Au/In/Ti (titanium/gold/indium/titanium) structures were used as the reflector and eutectic structures, respectively. After the wafer bonding process, the absorbing GaAs substrate was selectively removed in the H_2_O_2_:NH_3_ solution until the appearance of the GaInP layer. The GaInP ESL was eliminated in a HCl solution for 10 s. Bonded IR-LED wafers were sequentially cleaned with acetone and methanol to remove organic contamination, which was followed by removing the surface oxidation of the n-Al_0.2_Ga_0.8_As top window (front surface) and p-Si substrate (back surface) in a HF: DI (10:1) solution. After cleaning, the bonding pads were placed on the anterior and posterior surfaces using a combination of photolithography and selective etching. It is noteworthy that Au/AuGeNi (1000 nm/100 nm) on the anterior surface was deposited using an electron beam evaporator, and Au/Au/Be (500 nm/100 nm) was deposited on the posterior surface using a thermal evaporator. Figure 2 shows the structural schematic and compositional information of the reflective 850 nm IR-LED chip made with the Al/Au contact metal.

## 3. Results and Discussion

The reflective structures for high-power IR-LEDs were fabricated by combining the Si_3_N_4_ insulator and Ag reflector because the reflectivity of the Ag reflector was effectively sustained by the Si_3_N_4_ insulator. The Si_3_N_4_ insulator used is one of the important factors responsible for the increase of the series resistance in reflective IR-LEDs. To overcome this increased series resistance issue, the Si_3_N_4_ insulator was patterned to specific sizes and was filled with ohmic contact metal. Figure 3a shows the Si_3_N_4_ insulator patterned by ohmic contact metal and Figure 3b shows the corresponding optical microscopy images of patterned Si_3_N_4_ layer in the reflective IR-LEDs. In Figure 3a, the Si_3_N_4_ film was deposited on the top layer of the LED; it was then patterned periodically to form circular patterns using the lithographic and etching processes. The ohmic contact metal filled the patterns to an approximate thickness of 110 nm because of the thickness (110 nm) of the Si_3_N_4_ film. The Ag reflector and eutectic materials were sequentially deposited on the patterned Si_3_N_4_ film with ohmic contact metal. Herein, it was observed that circles with an approximate diameter of 10 μm were patterned every 20 μm in the Si_3_N_4_ insulator, as shown in the IR permeation image in Figure 3c. Figure 3d showed scanning electron microscopy images for the Ag reflector, eutectic structure, and patterned Si_3_N_4_/Al/Au in 850 nm IR-LED fabricated by wafer bonding process. Here, both eutectic structure and insulator in Figure 3d could easily be measured by the FIB-SEM system.

As mentioned earlier, the ohmic contact metal is a critical factor for the electric characteristics of the developed reflective IR-LEDs. Generally, Au/Be alloyed materials have been extensively used for reflective LEDs because use of the p-GaP top layer for reflective IR-LEDs. However, p-GaP is lattice-mismatched for AlGaAs materials used mainly for reflective 850 nm IR-LEDs. Furthermore, Be in Au/Be is not useful for AlGaAs-based quantum wells due to its high likelihood to be diffused to the active region [18,19].

On the other hand, here, the p-AlGaAs top layer was used for developed reflective IR-LEDs. Therefore, approximately 100 Al/Au patterns were inserted in one IR-LED chip (355 μm × 355 μm). The total area of patterns occupied almost 6% of the IR-LED chip.

Figure 4 shows the measurement scheme for patterned contact metal and current-voltage characteristics of patterned Au/Be for the p-GaP layer and those of the patterned Au/Be for the p-AlGaAs layer. The current-voltage characteristics of the patterned contact metals were systematically measured by a dual probe tip. Figure 4b shows the current-voltage characteristic for p-GaP/Au/Be and p-AlGaAs/Au/Be structures annealed at 550 °C. The current-voltage curve of the p-GaP/Au/Be structure demonstrates an excellent ohmic property. In the current-voltage curve of the p-GaP/Au/Be structure, the injection current was 100 mA, and the voltage was 5.75 V. Conversely, in the current-voltage curve of p-AlGaAs/Au/Be, at the same current conditions, the voltage was 3.44 V. Furthermore, p-AlGaAs/Au/Be yielded an abnormal curve compared with that of p-GaP/Au/Be. Given the fact that p-AlGaAs is the top layer of the IR-LED epi-wafer, Au/Be was not suitable for the development of reflective IR-LEDs.

To identify an optimum ohmic contact condition for the p-AlGaAs top layer, we investigated Al and the contained Au ohmic metal. The Al composition in the Al_x_Au_1−x_ alloy was controlled to 5%, 10%, and 20%.

Figure 5a shows the current-voltage curves for the pattern of Al_0.05_Au_0.95_, Al_0.1_Au_0.9_, Al_0.2_Au_0.8_, and conventional Au/Be for p-AlGaAs. For a current of 100 mA, the voltage was finely increased as the Al composition increased. The voltages of Al_0.05_Au_0.95_ and Al_0.1_Au_0.9_ were 3.59 V and 3.96 V, respectively, compared with that of the Au/Be (3.44 V). However, the voltage of Al_0.2_Au_0.8_ decreased to 3.13 V. The current-voltage curves show that the Al_x_Au_1−x_ (x = 0.05, 0.1) alloy has better ohmic characteristics than those of the Au/Be alloy for the p-AlGaAs layer. Furthermore, the current-voltage characteristics of Al_x_Au_1−x_ were obtained without any prior thermal treatments. The best ohmic characteristics were obtained by the patterned Al_0.1_Au_0.9_ alloy. From the current-voltage results of Figure 5a, it was found that the Al_0.1_Au_0.9_ is more suitable for the p-AlGaAs layer because it showed the highest voltage value when compared with the others at 100 mA. Based on the current-voltage results, the Al_0.5_Au_0.95_ and Al_0.2_Au_0.9_ alloyed materials have worse current-voltage characteristics than the Al_0.1_Au_0.9_ for p-AlGaAs layer.

Therefore, we tested different thermal conditions for p-AlGaAs/Al_0.1_Au_0.9_ structures. The injected thermal temperature was limited to 600 °C owing to the epitaxial growth temperature of 580 °C. It was found that the current-voltage characteristics increased as a function of thermal temperature. At a current of 100 mA, remarkable improved voltage outputs (5.8–6.0 V) were observed from 500 °C and 600 °C. The current-voltage characteristic (6.0 V) of the p-AlGaAs/Al_0.1_Au_0.9_ structure obtained at 500 °C yielded a higher voltage than that (5.75 V) of p-GaP/Au/Be at 550 °C. Therefore, it was demonstrated that the p-AlGaAs/Al_0.1_Au_0.9_ structure is essential to achieve excellent current-voltage characteristics and lower thermal conditions for reflective IR-LEDs.

For more detailed information, the component of the Al_x_Au_1−x_ alloyed material was investigated. To measure the composition ratio in the Al_0.1_Au_0.9_ alloyed materials, the Al_0.1_Au_0.9_ was melted and evaporated on the GaAs wafer. The Al_0.1_Au_0.9_ alloyed materials were measured by an energy dispersive spectrometer (EDS). It was found that deposited Al_0.1_Au_0.9_ film has significant smooth and clean surface. Figure 6b showed surface and EDS data of Al_0.1_Au_0.9_ alloyed materials. The result of EDS clearly presents that there is 9.8% Al and 90.2% Au deposited in Al_0.1_Au_0.9_ alloyed materials. We believed that appropriate ratio of Al_0.1_Au_0.9_ with a clean surface is very useful in improving ohmic characteristics for p-AlGaAs of reflective 850 nm IR-LEDs.

To obtain detailed information, the light power-current-voltage curve characteristics of the 850 nm reflective IR-LED chip which contained p-GaP/Au/Be, p-AlGaAs/Au/Be, and p-AlGaAs/Al_0.1_Au_0.9_ structures were also investigated. Herein, Au/Be-based and Al_0.1_Au_0.9_-based structures used for reflective IR-LED chips were thermally treated by post-annealing processes at 550 °C and 500 °C, respectively. Here, Au/Be and Al_0.1_Au_0.9_ have the same thickness because alloys are filled with a 110 nm-thick pattern hole of Si_3_N_4_ film. Normally, I–V property was not dependent on metal thickness, but rather the composition of either Be in the Au/Be alloy or Al in the Al/Au alloy.

Based on the current–voltage curves of Figure 7, the considerably increased forward voltage clearly proves that the AlGaAs/Au/Be structure was not suitable for the development of reflective IR-LEDs. Conversely, reflective IR-LED chips with p-AlGaAs/Al/Au structures yielded lower forward voltages compared with the reflective IR-LEDs with conventional p-GaP/Au/Be structures. This is owing to the reduction of series resistivity of the IR-LED chips caused by alloying aluminum (Al) and gold (Au). These results exhibited almost similar trends to those of the current-voltage characteristics in Figure 5.

From the light power-current curves at the injection current of 200 mA, it was found that the reflective IR-LED chip with the p-AlGaAs/Al_0.1_Au_0.9_ structure yielded the highest output power (182 mW), compared with the other reflective IR-LED chips. The output power of the IR-LED chip with the p-AlGaAs/Al_0.1_Au_0.9_ structure was approximately 65% higher than the output power (110 mW) of the IR-LED chip with the p-AlGaAs/Au/Be structure. An increase in the output power of the conventional reflective IR-LED with the p-GaP/Au/Be structure of 13.5% was observed in one p-AlGaAs/Al_0.1_Au_0.9_ structure. These results demonstrate that the electrical and optical characteristics of the reflective IR-LED could be improved effectively by using the alloyed metal Al_0.1_Au_0.9_.

Table 1 presents summarized data for light power-current-voltage characteristics of reflective IR-LED chips with different ohmic contact layers. From the current-voltage data in Table 1, it was found that the IR-LED chip with the p-AlGaAs/Al_0.1_Au_0.9_ structure showed lower forward voltage and higher output power when compared with the others under various current densities. In comparison with normal IR-LED chips with p-GaP/Au/Be especially, approximately 12.5% low forward voltage and 13% high output power were observed from one alloyed material with the p-AlGaAs/Al_0.1_Au_0.9_ structure. From the results of Table 1, it was demonstrated that Al_0.1_Au_0.9_ alloyed materials are essential to improve optical efficiency of reflective 850 nm IR-LEDs with a p-AlGaAs top layer.

At an injection current of 20 mA, the emission intensities of developed LED chips are compared in Figure 8. It can be seen that the emission intensity of the 850 nm IR-LED chip with the p-AlGaAs/Al_0.1_Au_0.9_ structure is approximately 1.23 times that of the conventional 850 nm IR-LED chip with p-GaP/Au/Be, which is equivalent to a 23 % increase. This could be attributed to the increase in the effective photon emission area, which is caused by the improvement of the ohmic characteristic of the p-AlGaAs/Al_0.1_Au_0.9_ structure. This indicates that the use of the Al_0.1_Au_0.9_ alloyed materials for the reflective 850 nm IR-LED with the p-AlGaAs top layer would significantly increase the optical efficiency of the device.

On the basis of these results, we can assume that using the patterned AlxAuy alloy effectively improves the series resistance of reflective IR-LEDs at lower thermal temperatures. It was verified that the I–V characteristics of the p-AlGaAs/Al_0.1_Au_0.9_ structure were more attractive than those of the p-GaP/Au/Be structure for reflective IR-LEDs at relatively low temperatures. As an extension of our results, we posit that the Al_0.1_Au_0.9_ alloy should be highly useful for application in reflective NIR or mid-infrared (MIR) LED at higher output powers.

## 4. Conclusions

In this study, the Al/Au alloy was investigated to improve the ohmic contacts for the top p-AlGaAs layers of reflective IR-LEDs. Based on the I–V characteristic measurements, the Al/Au alloy had higher values and improved ohmic curves when compared with the Au/Be alloy in the case of the p-AlGaAs layer. It was found that the Al_0.1_Au_0.9_ structure showed improved current voltage characteristics and ohmic curves at the thermal temperature of 500 °C. Based on all current voltage characteristics, the p-AlGaAs/Al_0.1_Au_0.9_ structure yielded the highest current voltage response (6.0 V) compared with that (5.75 V) of the p-GaP/Au/Be structure for reflective IR-LEDs. This result was justified by the light power- current- voltage results of reflective IR-LED chips with p-AlGaAs/Al_0.1_Au_0.9_, p-AlGaAs/Au/Be, and p-GaP/Au/Be structures. It was found that the lowest forward voltage of 1.56 V was obtained in the IR-LED chip case with the p-AlGaAs/Al_0.1_Au_0.9_ structure, and was 0.74 V and 0.18 V lower than those of the p-AlGaAs/Au/Be (2.3 V) and p-GaP/Au/Be (1.74 V) structures, respectively. The output power and I–V responses were similar. The reflective IR-LED chip with the p-AlGaAs/Al_0.1_Au_0.9_ structure yielded the highest output power (182 mW), which was higher than those of the reflective IR-LED chips which contained either p-GaP/Au/Be (161 mW) or P-AlGaAs/Au/Be (111 mW) structures. These results were attributed to its relatively lower forward voltage owing to the lower series resistance. As a result, it was verified that the Au/Be was not suitable for the p-AlGaAs layer. We infer that the AlGaAs/Al_0.1_Au_0.9_ structure is more useful than the conventional GaP/Au/Be structure for reflective IR-LED chips.

## Figures and Tables

**Figure 1 micromachines-14-01053-f001:**
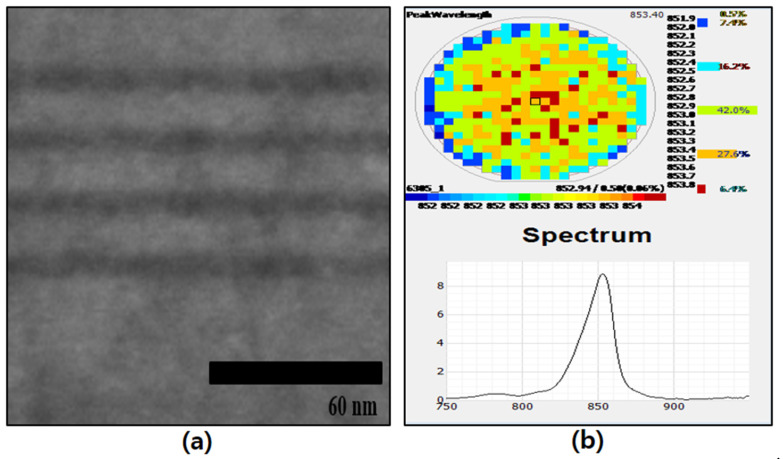
(**a**) TEM image and (**b**) Photo-luminescence for 850 nm MQW (active region) used for reflective 850 nm IR-LED.

**Figure 2 micromachines-14-01053-f002:**
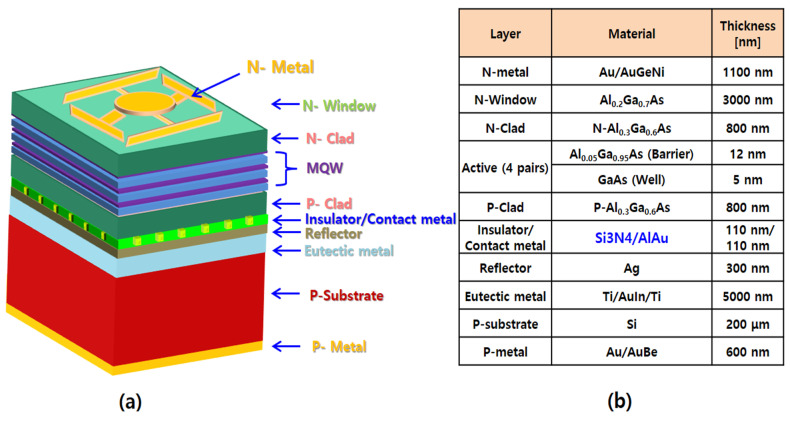
(**a**) Structural schematic and (**b**) composition information of reflective 850 nm IR-LED with Al/Au contact metal.

**Figure 3 micromachines-14-01053-f003:**
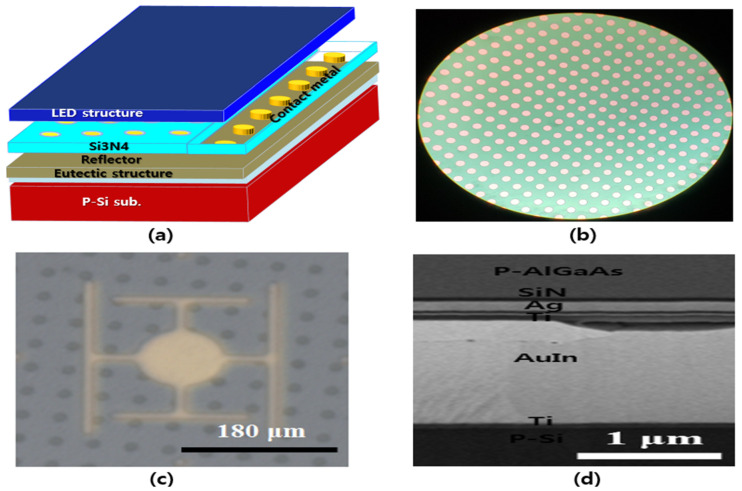
(**a**) Structural schematic, (**b**) optical microscopy surface image of Si_3_N_4_ insulator patterned by ohmic contact metal and corresponding (**c**) IR permeation surface image, and (**d**) cross-sectional scanning electron microscopy images of the reflective 850 nm IR-LED chip.

**Figure 4 micromachines-14-01053-f004:**
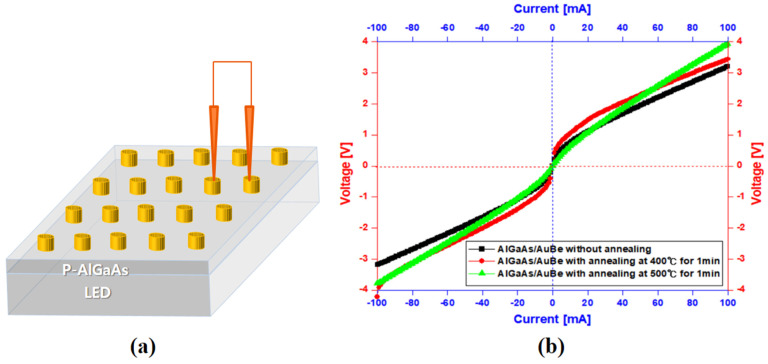
(**a**) Measurement scheme for patterned contact metal and (**b**) current-voltage characteristic of patterned Au/Be for p-GaP layer and patterned Au/Be for p-AlGaAs layer.

**Figure 5 micromachines-14-01053-f005:**
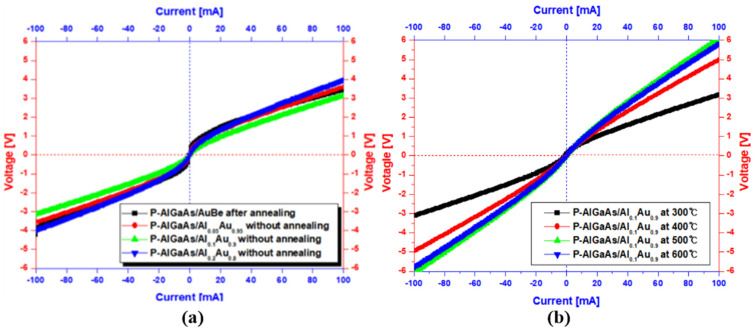
(**a**) Current-voltage characteristics of patterned Au/Be and Al_x_Au_1−x_ at various Al compositions for P-AlGaAs. (**b**) Current-voltage characteristics of patterned Al_x_Au_1−x_ at various Al compositions for P-AlGaAs annealed at different temperatures.

**Figure 6 micromachines-14-01053-f006:**
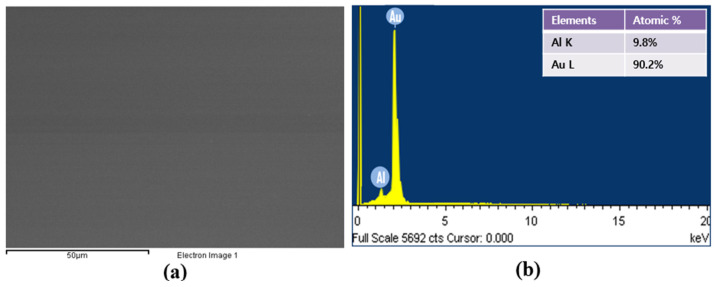
(**a**) Surface image and (**b**) EDS (energy dispersive spectrometer) data of Al_0.1_Au_0.9_ alloyed materials.

**Figure 7 micromachines-14-01053-f007:**
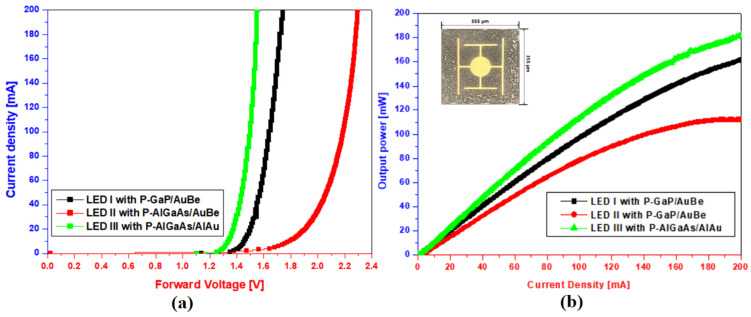
(**a**) Current–voltage and (**b**) current–light power for LED I (p-GaP/Au/Be), LED II (p-AlGaAs/Au/Be), and LED III (p-AlGaAs/Al/Au).

**Figure 8 micromachines-14-01053-f008:**
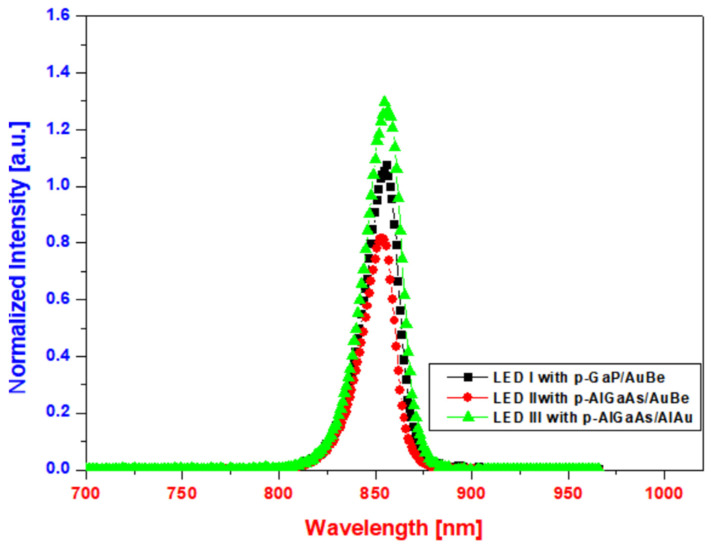
Electro-luminescence for LED I (p-GaP/Au/Be), LED II (p-AlGaAs/Au/Be), and LED III (p-AlGaAs/Al/Au).

**Table 1 micromachines-14-01053-t001:** Characteristic information of developed LEDs.

Decice	Contact Layer	Ohmic Metal for Contact Layer	Vf 2 [5 mA]	Vf 3 [100 mA]	Vf 4 [200 mA]	Po 1 [5 mA]	Po 2 [100 mA]	Po 3 [200 mA]
LED Ⅰ	P-GaP	AuBe	1.42 V	1.64 V	1.74 V	4.07 mW	96.49 mW	161.62 mW
LED Ⅱ	P-AlGaAs	AuBe	1.67 V	2.18 V	2.29 V	1.76 mW	77.94 mW	111.97 mW
LED Ⅲ	P-AlGaAs	Al_0.1_Au_0.9_	1.31 V	1.49 V	1.56 V	4.45 mW	113.75 mW	182.42 mW

Vf_Forward Voltage. Po_Output Power.

## Data Availability

The data that support the findings of this study was strictly limited. So please contact with corresponding authors.

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
