# Peer review of "Study on P-AlGaAs/Al/Au Ohmic Contact Characteristics for Improving Optoelectronic Response of Infrared Light-Emitting Device"

_micromachines, 2023, doi:10.3390/mi14051053_

Round 1

Reviewer 1 Report

In this article, authors present exciting results on the AlAu as an ohmic contact for IR LEDs. Authors have improved the performance by reducing the series resistance of the contacts. The article will be a good reference for researchers working on the topic. Reviewer recommends the publication of the article and has the following suggestions for its improvement.

1.      Current-voltage plots should be presented in the standard method. It is suggested to show current profiles on the y-axis and voltage profiles on the x-axis.

2.      Active area of the device should be given to the reader. It is suggested to provide the current density of the device.

3.      Table should be added to suggest the performance of the proposed device with reported devices. This table should highlight the merit of the presented device.

4.      Fig. 5 recommended LED performance, recommended to add the Electro luminance (EL) spectrum of different LEDs.

5.      If possible, please measure the work function of the different contacts for eg. by kelvin probe.

6.      Would be better to add a band diagram/schematic of the interface at the contacts.

7.      How the AlAu contacts in different ratios were deposited? by co-deposition? 

Author Response

The first reviewer point out;

In this article, authors present exciting results on the AlAu as an ohmic contact for IR LEDs. Authors have improved the performance by reducing the series resistance of the contacts. The article will be a good reference for researchers working on the topic. Reviewer recommends the publication of the article and has the following suggestions for its improvement

  1. Current-voltage plots should be presented in the standard method. It is suggested to show current profiles on the y-axis and voltage profiles on the x-axis.
  • We appreciate the comment from reviewer. Fig 7 (a) is revised.

  1. Active area of the device should be given to the reader. It is suggested to provide the current density of the device.
  • We appreciate the comment from reviewer. We inserted TEM image and P.L.(photo-luminescense) results of 850 MQW (active area) as figure 1. (Line no. 78 ~ 86)

  • Also, current density of the device was described in Fig. 7.

Figure 1. (a) TEM image and (b) Photo-luminescence for 850nm MQW (active region) used for reflective 850nm IR-LED.

  1. Table should be added to suggest the performance of the proposed device with reported devices. This table should highlight the merit of the presented device.
  • We appreciate the comment from reviewer. We have inserted the table 1 which presents characteristics of device. (Line no. 241 ~ 248)

  1. Fig. 5 recommended LED performance, recommended to add the Electro luminance (EL) spectrum of different LEDs.
  • We appreciate the comment from reviewer. The electro luminescence (E.L.) result was inserted as Fig. 8. (Line no. 251 ~258)

  1. If possible, please measure the work function of the different contacts for eg. by kelvin probe.
  • We appreciate the comment from reviewer. It is hard to measure the work function due to no proper equipment.

  1. Would be better to add a band diagram/schematic of the interface at the contacts.
  • We appreciate the comment from reviewer. We think that ohmic characteristics of either AuBe or AlAu alloyed metals for p-AlGaAs was well explained from the results of Fig. 4 and Fig. 5. Also, optical characteristics of 850nm IR-LED chip with either AuBe or AlAu alloyed materials was thoroughly described for reader in Fig. 7 and Fig. 8.
  • For whatever reason, we are sorry not to provides a lot of data for review.

  1. How the AlAu contacts in different ratios were deposited? by co-deposition? 
  • It was co-alloyed in-situ in evaporator system after mixing10 percent Al and 90 percent Au in crucible. The Al1Au0.9 alloyed layer deposited on wafer was measured by EDS analyzer. (Fig. 6, Line no 199 ~ 207)

Reviewer 2 Report

In this manuscript, the authors use Al/Au alloyed metal as the contact layer for improving the

optoelectric response of infrared light-emitting device. The lower series resistance between the contact metal and the top layer of p-AlGaAs results in the improved conductivity. The I–V characteristics of alloyed metals with different compositions were systematically characterized. Moreover, the annealing temperature was explored and the low temperature is benefit for the p-AlGaAs/ Al0.1Au0.9 structure to achieve high response. Finally, the highest output power was achieved by the Al/Au alloyed metal-composed devices. This manuscript needs some revisions, and some suggestions were listed below for the improvement of this manuscript.

1.      Why Al/Au alloyed metal demonstrated lower resistance, it is better to provide evidence and explain more in the manuscript.

2.      The experimental process for preparing patterned Si3N4 insulator and Al/Au alloyed metals should be provided in the manuscript.

3.      The experimental section about preparation of 850nm IR-LED should be detailedly provided.

4.      The structure about the Si3N4 insulator could not be seen clearly in Figure 2b and Figure 2c, and figure 2c is hardly understand, more information should be provide?The cross-section of Si3N4 insulator should be provided for comparison.

5.      The authors mentioned that “crystallization of p-AlGaAs/ Al0.1Au0.9” is helpful for the device, but nothing about crystallization is provided in the manuscript.

6.      The element mapping about the Al/Au alloy should be provided in the manuscript.

Author Response

The second reviewer point out;

In this manuscript, the authors use Al/Au alloyed metal as the contact layer for improving the optoelectric response of infrared light-emitting device. The lower series resistance between the contact metal and the top layer of p-AlGaAs results in the improved conductivity. The I–V characteristics of alloyed metals with different compositions were systematically characterized. Moreover, the annealing temperature was explored and the low temperature is benefit for the p-AlGaAs/ Al0.1Au0.9 structure to achieve high response. Finally, the highest output power was achieved by the Al/Au alloyed metal-composed devices. This manuscript needs some revisions, and some suggestions were listed below for the improvement of this manuscript.

  1. Why Al/Au alloyed metal demonstrated lower resistance, it is better to provide evidence and explain more in the manuscript.
  • We appreciate the comment from reviewer. In this manuscript, we would like to use p-AlGaAs contact layer to obtain 850 nm IR-LED with higher power due to the p-GaP contact layer is one of absorbing factors on lights emitted from active region. The GaP (5,45â„«) is lattice mismatched with AlxGa1-xAs(~5.62 â„«) mainly used for 850 nm IR-LEDs []. Therefore, there are a lot of lattice defects between GaP and AlGaAs materials. Lights in LED structure was absorbed and reduced by lattice defects. (Line no. 60 ~ 65)

[14] S. Hayashi, M. Nangu, T. Morikuni, S. Owa, N. S. Takahashi, “ Lattice-mismatched InGaP/GaAs(111)B liquid phase epitaxy with epitaxial lateral overgrowth”. Journal of Crystal Growth 311 (2009) 842.

  • Furthermore, figure 4 clearly showed that AuBe is not suitable for p-AlGaAs layer. On the other hands, it was found that the AlAs metal has better ohmic characteristic for p-AlGaAs layer in Fig 5. Therefore, we believe that these results support enough explains why AlAs alloyed metal is suitable for p-AlGaAs layer.

  1. The experimental process for preparing patterned Si3N4 insulator and Al/Au alloyed metals should be provided in the manuscript.
  • We appreciate the comment from reviewer. It is described in materials and methods section. (Line no. 96 ~ 103)

  1. The experimental section about preparation of 850nm IR-LED should be detailedly provided.
  • We appreciate the comment from reviewer. We inserted basic information for epitaxial growth process and its data of 850nm IR-LED. (Line no. 71 ~ 86)

  1. The structure about the Si3N4 insulator could not be seen clearly in Figure 2b and Figure 2c, and figure 2c is hardly understand, more information should be provide?The cross-section of Si3N4 insulator should be provided for comparison.
  • We appreciate the comment from reviewer. It was hard to express the Si3N4 layer in previous figure 3(b) due to its thickness is approximately 110 nm. Therefore, we inserted microscopy image (b), IR permeation image (c), and cross-sectional SEM image (c) in Fig. 3.
  • From Microscopy image(a) and cross-sectional SEML image(c), we try to show existence of Si3N4 film to reader. Also, we proved clearly existence of Si3N4/AlAu patterns in developed 850nm IR-LED chip in IR permeation image (b). (Line no. 132 ~ 143)

  1. The authors mentioned that “crystallization of p-AlGaAs/ Al0.1Au0.9” is helpful for the device, but nothing about crystallization is provided in the manuscript.
  • We appreciate the comment from reviewer. We omitted “crystallization” in the sentence because no evidence on crystallization.

  1. The element mapping about the Al/Au alloy should be provided in the manuscript.
  • We appreciate the comment from reviewer. We are sorry not to provides a lot of data for review. Here. we tried to provide EDS data on elements of Al/Au alloy material, which is inserted in Fig. 6.

Reviewer 3 Report

       The work suitable for the journal have found that AlAu is better than AuBe with distinct ohmic characteristic for p-AlGaAS based NIR-LEDs. But I don't think it can be accepted at the present form. Here are my comments.

1. The introduction is required to add more to explain why to select p-AlGaAS and what the former results for alloy layers necessary to show or support the work about AlAu interesting enough.

2. The authors need to explain why they use the only component of Al0.1Au0.9, and they also need to make clear what happen for other concentrations, better or worse?

3. What happen for the alloys annealed at different temperature for Fig. 3 and Fig.4? The authors need to have some discussion and more characterization, such as XRD and others to support the different I-V properties.

4. How the authors to have the real thickness for all of the layers shown in Fig.1? Please make the characterization part more detailed.

5. If the thickness of AuBe the same as that of AlAu for the comparison in Fig. 5? And what the effect of thickness on the I-V property?

6. What the phases is for the obtained alloys? The authors required to give the chemical analysis to support their components ( or make the characterization clearer in the revised manuscript), furthermore, to give a XRD characterization for thin film is better, if possible.

Author Response

The third reviewer point out;

The work suitable for the journal have found that AlAu is better than AuBe with distinct ohmic characteristic for p-AlGaAS based NIR-LEDs. But I don't think it can be accepted at the present form. Here are my comments.

  1. The introduction is required to add more to explain why to select p-AlGaAs and what the former results for alloy layers necessary to show or support the work about AlAu interesting enough.
  • We appreciate the comment from reviewer. In this manuscript, we would like to use p-AlGaAs contact layer to obtain 850 nm IR-LED with higher power due to the p-GaP contact layer is one of absorbing factors on lights emitted from active region. The GaP (5,45â„«) is lattice mismatched with AlxGa1-xAs(~5.62 â„«) mainly used for 850 nm IR-LEDs []. Therefore, there are a lot of lattice defects between GaP and AlGaAs materials. Lights in LED structure was absorbed and reduced by lattice defects. (Line no. 60 ~ 65)

[14] S. Hayashi, M. Nangu, T. Morikuni, S. Owa, N. S. Takahashi, “ Lattice-mismatched InGaP/GaAs(111)B liquid phase epitaxy with epitaxial lateral overgrowth”. Journal of Crystal Growth 311 (2009) 842.

  • Furthermore, figure 4 clearly showed that AuBe is not suitable for p-AlGaAs layer. On the other hands, it was found that the AlAs metal has better ohmic characteristic for p-AlGaAs layer in Fig 5. Therefore, we believe that these results support enough explains why AlAs alloyed metal is suitable for p-AlGaAs layer.

  1. The authors need to explain why they use the only component of Al0.1Au0.9, and they also need to make clear what happen for other concentrations, better or worse?
  • We appreciate the comment from reviewer. From I-V results of Fig. 5 (a). it was found that the Al0.1Au0.9 is more suitable for p-AlGaAs layer because it showed highest voltage value than the others at 100 mA. Base on I-V results, the Al0.5Au0.95 and Al0.2Au0.9 alloyed materials has worse I-V characteristic than Al0.1Au0.9 for p-AlGaAs layer.

For whatever reason, we are sorry not to provide data about either Al0.5Au0.95 or Al0.2Au0.9 alloyed materials annealed under various thermal temperature.

  1. What happen for the alloys annealed at different temperature for Fig. 3 and Fig.4? The authors need to have some discussion and more characterization, such as XRD and others to support the different I-V properties.
  • We appreciate the comment from reviewer. In Fig. 4. p-GaP/AuBe and P-AlGaAs/AuBe ohmic chacteristics were investigated. Here, higher temperature and longer annealing time are not useful for AuBe ohmic material, because it is well-diffused to active region [18-19].

  • From results of Fig 5 (b), the Al0.1Au0.9 alloyed was selected for p-AlGaAs layer, and then its I-V characteristic was investigated under various annealing temperatures.   

[18] A. Gaymann, M. Maier, K. Koher, Journal of Applied Physics 86 (1999) 4312-4315.  “Solid-solubility limits of Be in molecular beam epitaxy grown AlxGa1-xAs layers and short-period superlattices”. Journal of Applied Physics 86 (1999) 4312-4315. https://doi.org/10.1063/1.371362.

[19] G. E. Kohnke, M. W. Koch, C. E. C. Wood, G. W. Wicks, Appl. Phys. Lett. 66 (1995) 2786 ,“Beryllium diffusion in GaAs/AlGaAs single quantum well separate confinement heterostructure laser active regions” https://doi.org/10.1063/1.113475

  • For whatever reason, we are sorry not to provides a lot of data for review.

  1. How the authors to have the real thickness for all of the layers shown in Fig.1? Please make the characterization part more detailed.
  • We appreciate the comment from reviewer. Generally, epitaxial layer (Active region) was anaylized through using of TEM system in Fig 1(a). Either eutectic structure and insulator were easily measured by FIB-SEM system Fig 3 (d).

  1. If the thickness of AuBe the same as that of AlAu for the comparison in Fig. 5? And what the effect of thickness on the I-V property?
  • We appreciate the comment from reviewer. Both of them have the same thickness because alloyed metals are filled with 110nm thick pattern hole of Si3N4 film. Normally, I-V property was not depended on metal thickness, but composition of either Be in AuBe alloy or Al in AlAu alloy.

  1. What the phases is for the obtained alloys? The authors required to give the chemical analysis to support their components ( or make the characterization clearer in the revised manuscript), furthermore, to give a XRD characterization for thin film is better, if possible.
  • We appreciate the comment from reviewer. We are sorry not to provides a lot of data for review. Here. we tried to provide EDS data on elements of Al/Au alloy material, which is inserted in Fig. 6.

Round 2

Reviewer 2 Report

The quality of the revised manuscript has been improved and most of the questions have been satisfactorily addressed. The quality of the figures can be improved. I recommend it for publication in this journal after minor revision..

Author Response

The quality of the revised manuscript has been improved and most of the questions have been satisfactorily addressed. The quality of the figures can be improved. I recommend it for publication in this journal after minor revision

    • We appreciate the comment from reviewer. We improved figure, 1, 2, 4, 7, 8. in revised manuscript.

Reviewer 3 Report

Please add or combine the following answer to the comments in the revised manuscript. I think it not enough for the authors only have them to answer the reviewers, because they are also important to make the paper better for readers.

1. We appreciate the comment from reviewer. From I-V results of Fig. 5 (a). it was found that the Al0.1Au0.9 is more suitable for p-AlGaAs layer because it showed highest voltage value than the others at 100 mA. Base on I-V results, the Al0.5Au0.95 and Al0.2Au0.9 alloyed materials has worse I-V characteristic than Al0.1Au0.9 for p-AlGaAs layer.

2. We appreciate the comment from reviewer. Generally, epitaxial layer (Active region) was anaylized through using of TEM system in Fig 1(a). Either eutectic structure and insulator were easily measured by FIB-SEM system Fig 3 (d).

3. We appreciate the comment from reviewer. Both of them have the same thickness because alloyed metals are filled with 110nm thick pattern hole of Si3N4 film. Normally, I-V property was not depended on metal thickness, but composition of either Be in AuBe alloy or Al in AlAu alloy.

Author Response

Please add or combine the following answer to the comments in the revised manuscript. I think it not enough for the authors only have them to answer the reviewers, because they are also important to make the paper better for readers.

  1. We appreciate the comment from reviewer. From I-V results of Fig. 5 (a). it was found that the Al0.1Au0.9 is more suitable for p-AlGaAs layer because it showed highest voltage value than the others at 100 mA. Base on I-V results, the Al0.5Au0.95 and Al0.2Au0.9 alloyed materials has worse I-V characteristic than Al0.1Au0.9 for p-AlGaAs layer.
  • We appreciate the comment from reviewer. It was inserted. (Line 187 ~190)

  1. We appreciate the comment from reviewer. Generally, epitaxial layer (Active region) was anaylized through using of TEM system in Fig 1(a). Either eutectic structure and insulator were easily measured by FIB-SEM system Fig 3 (d).
  • We appreciate the comment from reviewer. It was inserted. (Line 85~86, Line 144 ~ 145 )

  1. We appreciate the comment from reviewer. Both of them have the same thickness because alloyed metals are filled with 110nm thick pattern hole of Si3N4 film. Normally, I-V property was not depended on metal thickness, but composition of either Be in AuBe alloy or Al in AlAu alloy.
  • We appreciate the comment from reviewer. It was inserted. ( Line 227 ~230 )
